# Observed mechanism for sustained glacier retreat and acceleration in response to ocean warming around Greenland

Evan Carnahan[1,2], Ginny Catania[1,3], and Timothy C. Bartholomaus[4]

[1]Institute for Geophysics, The University of Texas, Austin, TX, USA
[2]Oden Institute for Computational Engineering and Sciences, The University of Texas, Austin, TX, USA
[3]Department of Geological Sciences, The University of Texas, Austin, TX, USA
[4]Department of Geological Sciences, University of Idaho, Moscow, ID, USA

**Correspondence:** Evan Carnahan (evan.carnahan@utexas.edu)

**Abstract.** The dynamic loss of ice via outlet glaciers around the Greenland Ice Sheet is a major contributor to sea level rise. However, the retreat history and ensuing dynamic mass loss of neighboring glaciers are disparate, complicating projections of sea level rise. Here, we examine the stress balance evolution for three neighboring glaciers prior to, at the onset of, during, and, where possible, after retreat. We find no dynamic or thickness changes preceding retreat, implicating a retreat trigger at the ice-ocean boundary. Terminus retreat initiates large-scale changes in the stress state at the terminus. This includes a drop in along-flow resistance to driving stress followed by an increase in lateral drag and associated glacier acceleration. We find that the pre-retreat spatial pattern in stresses along-fjord may control retreat duration and thus the long-term dynamic response of a glacier to terminus retreat. Specifically, glaciers with large regions of low basal drag extending far inland from the terminus permit a chain of stress changes that results in sustained acceleration, increased mass loss, and continued retreat. Glaciers with similar basal stress conditions occur around Greenland. Our results suggest that for such glaciers dynamic mass loss can be sustained into the future despite a pause in ocean forcing.

## 1 Introduction

Currently, ice loss from the Greenland Ice Sheet (GrIS) is responsible for nearly a quarter of the total contributions to global sea level rise (IPCC, 2021) with up to two-thirds of mass loss from 1972 to 2018 from dynamic changes of marine-terminating outlet glaciers (Mouginot et al., 2019). Much of this dynamic change began in the mid to late 1990s when widespread terminus retreat began (Murray et al., 2015; Catania et al., 2018; Wood et al., 2021). The initiation of glacier retreat may occur through two mechanisms including an increase in terminus ablation at the ice-ocean boundary (Wood et al., 2021; Straneo and Heimbach, 2013) or a climate induced imbalance in ice flux arriving at the terminus - either due to thinning (Thomas and Bentley, 1978) or changes in glacier dynamics (Nick et al., 2009; van der Veen et al., 2011; Howat et al., 2010). While there is growing scientific consensus on the importance of increased ocean thermal forcing as the trigger of retreat in Greenland (Wood et al., 2021; Holland et al., 2008), glacier retreat and dynamic mass loss has continued, and in some cases accelerated, even after a widespread reduction in ocean thermal forcing (Wood et al., 2021). This indicates that factors other than ocean thermal forcing must control the evolution of multi-year retreat after the initial climate forcing. Furthermore, at the glacier scale, the ensuing

dynamic adjustment in response to terminus retreat results in heterogeneous changes in elevation and velocity (Felikson et al., 2017; Csatho et al., 2014; Moon et al., 2020). This variability means that, despite widespread observations of correlated retreat and glacier acceleration (King et al., 2020), we lack a physical understanding of the feedbacks between the different elements of glacier dynamic change.

Isolating the potential causes, and even the ordering, of terminus retreat and glacier dynamic changes requires frequent glacier and climate state information. Recent advances have filled in much of this state information for GrIS prior to, and through the onset of, retreat, ~1990s (Murray et al., 2015), to the present, e.g., Wood et al. (2021); Gardner et al. (2019); Mankoff et al. (2020); Morlighem et al. (2019); Goliber et al. (2022). However, elevation data with high-resolution coverage of narrow outlet glaciers is scarce prior to 2010 (Shean et al., 2016). This dearth of elevation data has restricted analysis of glacier dynamic evolution throughout retreat to a subset of glaciers in Greenland with long-standing observations, primarily Jakobshavn Isbrae (Thomas et al., 2003; Joughin et al., 2008; Amundson et al., 2010). However, even in regions with rich data, observations of stress fields prior to, and at the onset of, retreat do not include the terminus region (van der Veen et al., 2011), involve only one stress component, e.g., longitudinal coupling (Thomas, 2004), or require extensive interpolation of sparse spatio-temporal elevation data (Joughin et al., 2012). Studies attempting to circumvent elevation data scarcity augment observations with model simulations (Joughin et al., 2012; Bondzio et al., 2017). These studies reveal that a complex set of stress changes potentially link retreat and glacier dynamic changes (Bondzio et al., 2017), including the loss of longitudinal coupling resistance from melange breakup (Howat et al., 2010; Amundson et al., 2010) or a reduction in backstress (Nick et al., 2009), loss of basal resistance (Zwally et al., 2002), and decreases in lateral drag from shear margin weakening (van der Veen et al., 2011).

To elucidate the causal mechanisms for the onset, persistence, and possible cessation of outlet glacier retreat, it is necessary to have comprehensive observations that span transitions in glacier behavior. Here, we present such observations for three neighboring glaciers in Central Western Greenland with divergent retreat histories (Fig. 1) despite largely homogeneous oceanic and atmospheric forcing (Felikson et al., 2017; Wood et al., 2021). To fill in the gap in elevation data during the onset of retreat, we use an image processing pipeline (Girod et al., 2017) to remove systematic errors in ASTER imagery (Fujisada et al., 2005), enabling a new time series of Digital Elevation Models (DEMs) for GrIS outlet glaciers during a critical time period in their evolution. We produce DEMs and integrate them with velocity and bed elevation datasets to estimate the stresses controlling glacier flow through inverse methods (MacAyeal, 1992; van der Veen and Whillans, 1989). We use this comprehensive multi-glacier dataset of stress, elevation, and velocity to investigate the potential forcing mechanism responsible for retreat, and to determine why the glacier dynamic response to retreat diverges across time and space.

## 2  Methods

Our region of interest includes three neighboring outlet glaciers in Central Western Greenland with divergent histories (Fig. 1). The terminus of Ingia Isbrae was stable from 1985 to 2002, after which it began a steady retreat of over 8 km, continuing through 2021. Umiamako Isbrae, began retreating a year earlier, in 2001. Unlike Ingia, Umiamako retreated only 4 km before

abruptly restabilizing in 2009, where the terminus has remained since. Finally, Rink Isbrae may have retreated from 1998 to 2010, however no secular trend emerges from seasonal fluctuations (Catania et al., 2018). Rink also notably has a floating terminus and an ice flux that is nearly 10 times larger than neighboring Ingia and Umiamako (Catania et al., 2018).

We use time series elevation and velocity data to calculate the evolving stress fields for each glacier in this region using the force balance method following van der Veen and Whillans (1989). This method provides snapshots of the glacier stress state in time and is therefore useful to examine how the stress state varies during changes in terminus position. The force balance method has previously been used in Greenland to understand the behavior of individual glaciers (van der Veen et al., 2011) or multiple glaciers during single time periods (Bartholomaus et al., 2016; Stearns and van der Veen, 2018; van der Veen

et al., 2011; Enderlin et al., 2016; Meierbachtol et al., 2016), as well as in Antarctica (Price et al., 2002; Stearns et al., 2005) and Alaska (O'Neel, 2005; Enderlin et al., 2018). The force balance method assumes that the driving stress, $\tau_d$, is supported by basal drag, $\tau_b$, and depth-integrated longitudinal coupling and lateral drag, $F_{\mathrm{lon}}$ and $F_{\mathrm{lat}}$, respectively. We choose a sign convention where positive values of driving stress act in the direction of flow and positive values of all other stresses oppose flow,

$$\tau_b = \tau_d - F_{\mathrm{lat}} - F_{\mathrm{lon}}. \tag{1}$$

The depth-integrated components in the x-direction are given by

$$F_{\mathrm{lat}} = -\frac{\partial}{\partial y}(HR_{\mathrm{xy}}) \quad \text{and} \quad F_{\mathrm{lon}} = -\frac{\partial}{\partial x}(HR_{\mathrm{xx}}), \tag{2}$$

where $H$ is the ice thickness and the resistive stresses, neglecting vertical shearing, are given by

$$R_{xx} = B\dot{\varepsilon}_e^{1/n-1}(2\dot{\varepsilon}_{xx} + \dot{\varepsilon}_{yy}), \tag{3}$$

$$R_{yy} = B\dot{\varepsilon}_e^{1/n-1}(\dot{\varepsilon}_{xx} + 2\dot{\varepsilon}_{yy}), \tag{4}$$

$$R_{xy} = B\dot{\varepsilon}_e^{1/n-1}\dot{\varepsilon}_{xy}, \tag{5}$$

where $B$ is the viscosity rate factor, $n$ is the stress exponent in Glen's Flow Law, $\dot{\varepsilon}_{ij}$ is the depth-averaged strain rate, and the effective strain rate is given by

$$\dot{\varepsilon}_e = (\dot{\varepsilon}_{xx}^2 + \dot{\varepsilon}_{yy}^2 + \dot{\varepsilon}_{xx}\dot{\varepsilon}_{yy} + \dot{\varepsilon}_{xy}^2)^{1/2}. \tag{6}$$

Here, we assume that ice flow does not very with depth, i.e., we neglect vertical shearing, so the depth-averaged strain rate is equal to the surface strain rate. Depth-integrated resistive stresses are estimated with a stress exponent of $n = 3$ and a viscosity

rate factor of $B = 300$ kPa yr$^{1/3}$, similar to that used for nearby Jakobshavn Isbrae (van der Veen et al., 2011). The driving stress in the x-direction takes the form,

$$\tau_d = -\rho g H \frac{\partial h}{\partial x} \tag{7}$$

where $\rho$ is the density of ice, $g$ is the gravitational driving stress, and $h$ is the glacier surface elevation. Basal drag is calculated as the residual of the depth-integrated stresses and driving stress, Eq. (1). We calculate the force balance in 2D, and each stress

component is then oriented into an along-flow across-flow coordinate system. Results are shown for the along-flow coordinate
system. A thorough overview of the force balance method is given in van der Veen, C.J. (2013).

The calculation of stresses requires glacier-wide observations of velocity, bed, and surface elevation at coincident time periods (van der Veen and Whillans, 1989; MacAyeal, 1992). Such observations are integrated into a model of glacier flow that can range in complexity from shallow-shelf approximations (van der Veen and Whillans, 1989) to time-dependent 2D flow (Goldberg et al., 2015) to 3D inversions (Shapero et al., 2016). We assume a simple shallow-shelf approximation with basal
drag, Eq. (1) (MacAyeal, 1989), which is appropriate for our study area as basal sliding likely dominates viscous deformation (Bartholomaus et al., 2016). Such simplicity is particularly useful considering the longstanding debate on an appropriate relation between basal drag and sliding speed for fast-flowing outlet glaciers in Greenland (Stearns and van der Veen, 2018; Kamb, 1970; Nye, 1970; Lliboutry, 1979; Zoet and Iverson, 2020; Joughin et al., 2019), which more complex models must assume.

We use a range of data sets to piece together surface elevation changes for this region. Prior to the onset of retreat, the best available DEM comes from historical air photos taken in 1985 (Korsgaard et al., 2016). We use ASTER stereo pairs and the MicMac processing scheme (Girod et al., 2017) to produce improved ASTER DEMs for ∼2002 (Ingia: June 2003, Umiamako: July 2002, Rink: July 2001), which neighbors the retreat onset for Ingia and Umiamako. Finally, we use the GIMP DEM with a nominal year of 2007 (data from 2003 - 2009) and DigitalGlobe DEMs from ∼2015 (data from 2012 - 2015 for our region)
(Howat et al., 2014), which provide continuous glacier coverage and surface slopes. We use a land mask (Howat et al., 2014) to mask stable terrain for coregistration of DEMs (Nuth and Kääb, 2011) and the 1985 DEM as a baseline reference. We find random errors over stable terrain with root mean squared errors less than 10 m for the corrected ASTER imagery, comparable to the agreement we find between the other (non-ASTER) co-registered DEMs of ∼15 m.

We consider surface elevation only within the up-glacier confined outlet glacier trough, shown in Figure 1, due to the limits
of the 1985 DEM and the dominance of sliding within this fast flowing region, which the force balance method assumes. The down glacier extent of surface elevation DEMs is cropped using the July terminus location for each epoch examined (Catania et al., 2018). We evaluate the surface elevation change in this region using DEM differencing for all glaciers prior to, at the onset of, during, and after retreat. We further calculate the amount of thinning prior to retreat resulting from changing ice dynamics alone by removing the surface mass balance anomaly, integrated over the roughly 15 years preceding retreat, from
total thinning (Felikson et al., 2017). This allows us to isolate the source of surface elevation changes that may have occurred pre-retreat. Surface mass balance is provided from the regional climate model RACMO2.3p2, downscaled to 1 km (Noël et al., 2018).

Two additional data products are necessary for the force balance method: velocity data at each epoch and bed elevation. We use annual mean surface velocity mosaics derived from Landsat imagery for each year in our study period (Gardner et al.,
2019). We remove and interpolate velocity data with low annual scene pair counts and high errors ($< 1\%$ of points on each glacier). During 2002 several of the points removed are clustered in the near-terminus region of Umiamako, which results in a slight apparent decrease in velocity. Bed elevation is assumed to be invariant throughout the study period and given by BedMachineV3 (Morlighem et al., 2017). The 1985 surface elevation DEM is posted at a 25 m resolution, whereas all other

DEMs have a 30 m resolution. Velocity data has a grid resolution of 240 m and the bed topography has 150 m resolution. All

data products are smoothed to a regular 250 m grid using a 500 m Gaussian kernel, which both allows for collocation of data points and helps to reduce errors in input data that can impact stress estimation (Meierbachtol et al., 2016; O'Neel, 2005).

Calculated stresses are smoothed with a 1 km Gaussian kernel, and are shown for the centerline of each glacier using centerlines derived by Felikson et al. (2017). This final smoothing follows previous studies, and is necessary for physical interpretation of stresses, as calculated stresses can not be interpreted at length scales below the stress coupling length of each

glacier (O'Neel, 2005; Bartholomaus et al., 2016; Meierbachtol et al., 2016; Enderlin et al., 2016; Stearns and van der Veen, 2018). For the glaciers in our study region, the stress coupling length is at a minimum ∼2 km, i.e., the length across the kernel used for smoothing. Furthermore, quantitative results presented here for stress changes are averaged over a stress coupling length, which is calculated individually for each glacier as four times the ice thickness in the terminus region (Enderlin et al., 2016). We define the area within one stress coupling length of the terminus as the 'near-terminus' region.

Errors in calculated stresses arise from errors in all input datasets, i.e., bed topography, surface DEMs, and velocity data. Bed topography is assumed to be invariant in time, thus errors associated with bed topography are similarly consistent. Errors in inferred basal drag using the force balance with BedMachineV3 are estimated to be <15 kPa (Stearns and van der Veen, 2018). As we are largely concerned with year-to-year changes in stress we ignore uncertainty from errors in bed elevation. Errors in surface DEMs and velocity data result in relative errors in stress between study years. We ignore errors in the surface

DEM as velocity errors make up the vast majority of uncertainty in stress estimation (O'Neel, 2005; van der Veen, C.J., 2013). We analytically propagate errors in velocity through our model following Taylor (1996) and van der Veen, C.J. (2013). We assume errors in annual velocity maps and stress fields arise independently, i.e., spatial averaging reduces error. It is important to note that the values given for velocity errors "allow for the formal propagation of errors" but "provided errors... should be used as qualitative metrics for assessing errors" (Gardner et al., 2019). As a result, the formal propagation of errors through

our model should be taken as a similarly qualitative metric. The resulting uncertainty for each glacier, year, and stress term is given in the supplementary materials. When the changes in stress we observe are on the order of uncertainty, the result will be discussed in context of that uncertainty.

On average, we find comparable uncertainty in stress terms to previous studies in Greenland, e.g., Enderlin et al. (2018); Stearns and van der Veen (2018). However, during certain years we find some regions of very high uncertainty due to small

regions of coincident low strain rates and high velocity errors (Supplementary Figs. 1-3). We also observe some locations in the near-terminus region with negative basal drag. After averaging over a stress coupling length, values of negative basal drag that exceed uncertainty only occur for Ingia in 2015. Such values of negative basal drag are not physical and are likely related to our model's assumption of no vertical deformation (Enderlin et al., 2016) or a temporally invariant viscosity rate factor (van der Veen et al., 2011). One possible solution is to adjust the viscosity rate factor such that basal drag remains above zero for all

150    years (van der Veen et al., 2011). However, changing the rate factor only scales year-to-year changes in stress, and since we are largely concerned with relative, temporal, changes in stress components during retreat, such a scaling would not fundamentally change our findings. Finally, the region of negative basal drag and those that are zero within error, are largely restricted to

the near-terminus region, where all glaciers approach flotation and near-zero values of basal drag are expected, providing an independent source of validation for the rate factor used (Fig. 1).

## 3 Results

The three glaciers exhibit distinctly different force balance configurations and a high degree of spatial variability along flow (Fig. 2, 3, and 4). Excluding the regions within ∼5 km of the glacier termini, all glacier stress profiles maintain consistent spatial patterns along-flow over the 30-year observation period. Average per point absolute changes in inferred basal drag above the near-terminus region along the centerline of each glacier between each epoch are 11 kPa for Ingia (Fig. 2), 25 kPa for Umiamako (Fig. 3), and 16 kPa for Rink (Fig. 4). Estimated average error for basal drag along the centerline of each glacier is 43 kPa, 69 kPa, and 29 kPa for Ingia, Umiamako, and Rink, respectively. Notably, each of these changes in basal drag are potentially zero within error for all epochs. This lack of up-glacier stress changes for both retreating and stable glaciers implies that changes in stresses in the near-terminus region are primarily responsible for the pronounced acceleration observed during retreat. Thus, we focus our analysis on changes that occur in this near-terminus region. This region is identified by the horizontal bars in Figs. 2, 3, and 4. Near-terminus stress changes predominantly occur after the onset of retreat (Fig. 2 and 3). Coupled with the lack of dynamic changes in glaciers not experiencing retreat (Fig. 4), this implies that in the absence of retreat secular glacier dynamics are largely invariant for these three glaciers during our study period.

### 3.1 Pre-retreat

All three glaciers approach flotation at their termini pre-retreat as shown by height above buoyancy approaching zero (Fig. 1e). For Umiamako pre-retreat, height above buoyancy rapidly increases inland of the terminus, whereas Ingia's pre-retreat height above buoyancy remains within 200 meters of flotation for 10s of kilometers inland (Fig. 1e). A portion of the Rink terminus (up to 3 km) remains floating throughout the observational period.

We examine changes in ice thickness and glacier dynamics prior to retreat to deduce potential retreat mechanisms. Climate-induced changes in ice flux to the terminus can cause terminus retreat and may result from several mechanisms including 1) shear margin weakening through the release of latent heat due to melt-refreezing in crevasses (van der Veen et al., 2011); 2) reduction in bed resistance due to enhanced basal slip (Zwally et al., 2002); 3) decreases in terminus backstress forced by ice-ocean interactions that causes acceleration and subsequent retreat (Nick et al., 2009); 4) or enhanced surface melt leading to thinning induced retreat (Thomas and Bentley, 1978). In all of these cases, we would anticipate observations of significant surface elevation lowering or stress changes prior to retreat. Similar to 3), but in the reverse order, the climate system can also force retreat through increased frontal ablation and successive dynamic changes (Motyka et al., 2011). In this case, we would not expect to observe surface lowering and/or dynamic changes prior to retreat.

We examine total ice thickness changes and ice thickness changes due to ice dynamics alone. Prior, and up to the onset of its retreat in 2002, Ingia experienced a small, near-uniform, thickening of $5 \pm 6$ m across its trunk from 1985 to 2003 with 11 m of thickening resulting from ice dynamics alone (Fig. 1). Similar to Ingia, Umiamako experienced small thickness changes up

to the onset of retreat ($<10 \pm 14$ m of near-uniform total thinning, 5 m of dynamic thinning from 1985 to 2002). Although, this thinning is small and potentially zero, it is still possible that this thinning may have contributed to the retreat of Umiamako. We follow Wood et al. (2021) and calculate the amount of retreat induced by thinning alone on Umiamako. We find that only 0.5 meters of retreat from 1985 to 2003 can be explained by thinning. Negligible rates of dynamic thickness change are consistent with observed negligible changes in inland ice flux/velocity and minor changes in all components of the stress balance prior

to retreat for both glaciers (Fig. 2 and 3). The lack of changing ice flux and thinning prior to retreat thus implicates ice-ocean calving front processes as being primarily responsible for the retreat of Ingia and Umiamako.

## 3.2 Dynamics of retreating glaciers

During the initial stage of retreat, Ingia and Umiamako experience a drop in the degree to which along-flow longitudinal stresses support driving stress. For Ingia (retreat onset in 2002), we observe a drop in longitudinal coupling resistance in the

195 near-terminus region of 10 kPa from 2003 to 2007, representing a 20% reduction in overall resistance to driving stress (Fig. 2e). Conversely, the Umiamako near-terminus (retreat onset in 2001) experiences an increase in longitudinal coupling between 1985 and 2007 of 20 kPa, however driving stress substantially increases over this same time period (Fig. 3b and e). As a result of this driving stress increase, the proportion of Umiamako's driving stress resisting longitudinal coupling actually drops by ~16% at the terminus between 1985 and 2007, a comparable drop to what we observe at Ingia. Such a reduction in the

200 percent of driving stress supported by longitudinal coupling occurs largely because as these glaciers retreat, they enter parts of the fjord where longitudinal coupling is lower. For both glaciers, the along-flow pattern of longitudinal coupling decreases upglacier from the terminus. The changes in longitudinal coupling in the near-terminus region that we observe are on the order of estimated uncertainty in stress. For Umiamako though, the decrease in longitudinal coupling resistance to driving stress is due to an increase in driving stress and thus the reduction in fractional resistance to driving stress is well outside uncertainty.

For Ingia, the decrease in longitudinal coupling from pre-retreat to during retreat, 2007, is present for both pre-retreat years, 1985 and 2002. Furthermore, the pattern of decreasing longitudinal coupling inland of the terminus, which drives the decrease in resistance during retreat, is visible in all three years. These points provide evidence that the observed decrease in longitudinal coupling resistance to driving stress at Ingia is real, however the magnitude of the decrease is uncertain.

During the next phase of retreat the stress state at the terminus begins to differ from the spatial pattern that existed prior to

210 retreat, and each glacier experiences large changes in velocity that occur shortly after the onset of retreat. The Ingia terminus region does not experience a significant change in driving stress as the terminus retreats (Fig. 2a and b). Instead, the drop in terminus backstress on Ingia is followed by an almost three-fold increase in lateral drag (30 to 85 kPa) and a commensurate drop in basal resistance from 2007 to 2015 (Fig. 2d). This increase in lateral drag results from large temporal increases in across-glacier gradients in velocity, as Ingia undergoes a 56% increase in centerline velocity. The Umiamako terminus also

experiences a major increase in lateral drag in response to retreat (from 24 kPa in 1985 to 162 kPa in 2007) associated with a ~50% increase in centerline velocity (Fig. 3). However, for Umiamako, and in contrast to Ingia, all resistive stresses increase as the terminus retreats, not only lateral drag.

### 3.3 Dynamics of abbreviated versus sustained retreat

The stress balance at the terminus can change due to the retreat of the terminus to locations along-fjord where spatial patterns in stress are variable (determined by the pre-retreat geometry). In addition, the stress balance at the terminus can change because of temporal changes in stress at a fixed location. For Umiamako, we find that the near-terminus region experiences pronounced increases in driving stress, lateral drag, and basal drag, largely because the terminus retreats to locations along fjord where these values were high pre-retreat, thus these apparent increases in stresses are not due to temporal increases at fixed locations. For example, during the retreat of Umiamako from 1985 to 2007, near-terminus driving stresses increase by nearly 300 kPa, but the vast majority (260 kPa) of this increase arises because of an along-fjord increase in stress between 0 and 4 km, with only a fraction of the total (40 kPa) due to a temporal increase in driving stress at 4 km (Fig. 3b). Umiamako experiences a similar magnitude increase in near-terminus basal drag, also due to along-fjord variability, and stabilizes at a bed high, ∼4 km, at a pre-retreat fjord maximum in basal drag (Fig. 3c). After the terminus stabilizes, Umiamako continues to thin from 2007 to 2015 (Fig. 1c) with resulting decreases in near-terminus driving stress (∼30%) and all resistive stresses (Fig. 3). Importantly, basal drag remains high within a coupling length of the stable 2015 terminus position, possibly explaining the persistence in terminus position at this location.

Unlike Umiamako, Ingia has very little along-fjord variability in stress components within the first 15 km of its terminus (Fig. 2), and maintains persistently low basal drag at the terminus throughout its retreat. The spatially prevalent low-drag conditions throughout the terminus of Ingia are similar to ice shelf conditions. Because the bed cannot support an increase in stress, any reduction in backstress during retreat must be compensated by an increase in lateral drag. Indeed, subsequent to its retreat and drop in longitudinal backstress, Ingia experiences an acceleration in ice flow velocity and associated two-fold temporal increase in lateral drag (Fig. 2a and d). These results demonstrate that the unique pre-retreat along flow variability in stress state, largely a reflection of glacier geometry, dictates the dynamic changes that ensue after a period of terminus retreat.

### 3.4 Dynamics of stable glaciers

Unlike Umiamako and Ingia, Rink Isbrae does not experience a significant retreat over the observational period. Rink has nearly double the driving stress, velocity, and thickness of Ingia and Umiamako, and delivers a much larger ice flux to the terminus (Fig. 4). This high ice flux drains a large inland ice catchment (Mouginot et al., 2019) and results in high resistance to driving stress from longitudinal stresses as flow is funnelled through the outlet trough (∼31% of driving stress, Fig. 4e). The large contribution of longitudinal resistance to driving stress is unique to Rink; Ingia and Umiamako have longitudinal resistance that is just 4% of driving stress on average upstream of the near-terminus region. Furthermore, for two kilometers behind the near-terminus region of Rink, the fraction of longitudinal coupling supporting driving stress stays roughly the same (increases by ∼2% when averaged over the stress coupling length, Fig. 4e). As a result, and opposite to Umiamako and Ingia, a small terminus retreat on Rink would not result in a drop in near-terminus backstress. It is worth noting that once Umiamako restabilized in 2009, it exhibits an along-flow pattern in the fractional resistance from longitudinal coupling to driving stress

that is close to that found on Rink (Fig. 3e), while Ingia maintains a pattern of decreasing longitudinal resistance inland from the terminus through 2015 (Fig. 2e)

## 4   Discussion

Recent work by Wood et al. (2021) shows that ocean warming has likely enhanced terminus melt and melt-induced calving at hundreds of glaciers across Greenland, inducing widespread retreat. Their analysis of Umiamako and Ingia suggests that the
255 termini of these glaciers have easier access to warm ocean waters compared to Rink, which sits on a protective submarine ridge. Our results are in good agreement with their observations. We show a lack of dynamic change and thinning prior to the retreat of both Ingia and Umiamako, which implicates ocean forcing as the driver for retreat. While an exhaustive study, Wood et al. (2021) were not able to attribute the retreat mechanism for 87 glaciers due to the lack of ocean thermal and bathymetry data within these fjords, which are required for their methods. Our results suggest that an examination of comprehensive glacier
elevation, velocity, and surface mass balance data at the onset of retreat provides an independent measure of retreat attribution that can be assessed regardless of ocean data availability.

Through the examination of the time-varying stress state after the onset of retreat, we observe a consistent reduction in longitudinal resistance to driving stress for both Umiamako and Ingia, which is followed by increases in lateral drag and associated acceleration during retreat; although their is uncertainty in the exact magnitude of this drop in longitudinal backstress, our esti-
265 mates suggest it amounts to roughly a halving of its role in resisting driving stress. Such observations agree well with previous modeling of dynamic changes during the retreat of outlet glaciers around Greenland, which indicate longitudinal backstress reductions are the initial dynamic change during retreat (Nick et al., 2009; Bondzio et al., 2017). These observation suggest that one potential mechanism for the widely observed acceleration of outlet glaciers around Greenland (Moon et al., 2020; King et al., 2018, 2020) is a response to coupled changes in lateral drag and near-terminus longitudinal backstress initiated by
270 terminus retreat. However, glaciers around Greenland inhabit a wide range of geometries, climate regimes, and fjord geometries (Morlighem et al., 2017; Catania et al., 2018; Felikson et al., 2021) so future study is likely necessary to understand the prevalence of this proposed dynamic connection between retreat and acceleration.

Our results provide insight into how long-term retreat and dynamic change can continue even after ocean thermal forcing decreased in 2008 (Wood et al., 2021). We find that retreat persists due to fjord-specific patterns in stress state that are set
by the glacier geometry. For some glaciers, like Ingia, the fjord geometry permits low basal drag extending far inland. This region of low basal drag occurs where the submarine bed topography is shallowly retrograde extending far inland up to 15 km (Fig. 2a). As a result of this topography, as Ingia retreats the near-terminus region continuously experiences little resistance from basal drag, and the post-retreat drop in backstress is compensated by a large temporal increase in lateral drag and high across-glacier gradients in velocity. Glacier acceleration increases ice discharge, further exacerbating the initial retreat, leading
to prolonged mass loss that continues beyond the initial climate forcing. In contrast, Umiamako has a highly variable spatial pattern in along-fjord basal drag that is conducive to stabilization against continued retreat. A pre-retreat basal drag maximum is maintained throughout retreat four kilometers behind the 1985 terminus, where the bed topography shallows. The initial drop

in backstress for Umiamako is thus followed by an abbreviated increase in lateral drag and velocity before restabilization at the basal drag maximum. The role of glacier geometry in dictating retreat length post-climate perturbation is documented both theoretically (Schoof, 2007) and through observations (Catania et al., 2018). We further find a strong role for glacier geometry in setting the terminus stress state and ensuing dynamic changes throughout retreat, both during and after climate forcing.

Beyond retreat duration, glacier geometry has been shown to control the amount of thinning that occurs after a terminus perturbation (Felikson et al., 2017, 2021). Felikson et al. (2017) identifies points in the along flow glacier domain where a thinning wave initiated by terminus retreat will diffuse to, i.e., the Peclet limit. The Peclet limit occurs at 15 kilometers for Ingia but outside the area covered by DEMs for Umiamako and Rink (>45 km inland). For Ingia, the flow regime from the terminus to the Peclet limit is characterized by low driving stress and basal drag (Fig. 2b and c). At the Peclet limit both driving stress and basal drag increase by nearly a factor of two. This suggests that the ability for thinning waves to diffuse up glacier is linked to the stress state of the glacier, and potentially to the ability for stress changes to be transferred upstream (Bondzio et al., 2017). Fundamentally, both the pre-retreat stress state that we identify and the Peclet thinning limit identified by Felikson et al. (2017) highlight the importance of glacier geometry in determining the dynamic response to retreat. Furthermore, we find here that thinning is subsequent and in response to retreat. This ordering is consistent with, and helps to fill in, the chain of events suggested by Felikson et al. (2017).

We find a strong imprint of geometry on the observed basal drag, e.g., the first 15 km of Ingia (Fig. 2a and c). This primacy of geometry is largely due to the long time scales over which we analyse changes in stress, our focus on relative temporal changes, and our analysis of an area that is likely dominated by sliding over soft beds (Andrews et al., 1994). However, subglacial hydrology and bed materials play a strong role in regulating basal friction as well, e.g., Joughin et al. (2019); Schoof (2005); Zoet and Iverson (2020); Zwally et al. (2002). There are clear examples where changes in subglacial hydrology are almost wholly responsible for basal drag changes. For example, on seasonal timescales the glacier geometry is marginally different and yet glacier dynamics show substantial variability that is best explained by changes in subglacial hydrology (Howat et al., 2010). Such changes are far below the temporal resolution of our analysis, but provide a potential avenue for future work to decipher where and when geometric or subglacial processes are paramount in setting basal drag.

The catastrophic retreat of Ingia mirrors two other glaciers with observed or modeled stress fields: Columbia Glacier in Alaska and Sermeq Kujalleq (also known as Jakobshavn Isbrae) in Greenland. On Columbia, locations of maximum basal drag are located far upstream, ∼10 km from the pre-retreat terminus position, and the terminus region is characterized by low basal drag throughout retreat, as it approaches flotation (O'Neel, 2005). During retreat lateral drag rapidly increases, resulting in a four fold increase in terminus lateral drag from 1980 to 2005 (O'Neel, 2005). On Jakobshavn, basal drag supports only a small fraction of the driving stress, likely due to a weak bed and low effective pressures (Shapero et al., 2016). As a result of these low drag conditions, changes in terminus backstress initiated during retreat are rapidly transmitted upstream (Bondzio et al., 2017; Joughin et al., 2012). The contrasting dynamics between Umiamako and Ingia, along with the agreement between Ingia, Columbia, and Jakobshavn, suggest that the size of the region of low basal resistance upstream of the terminus preconditions the magnitude of glacier retreat and dynamic change in response to climate forcing. Furthermore, and similar to Umiamako, the retreat of Columbia slowed at a constriction with an along-fjord maximum in basal drag (O'Neel, 2005), indicating that such

locations may present points of stability during, or after, climate forced retreat. Conditions of low basal drag throughout the near-terminus region of glaciers are not restricted to the glaciers in this study, but occur around Greenland (Shapero et al., 2016; Sergienko et al., 2014; Seddik et al., 2018; Bartholomaus et al., 2016; Nick et al., 2009; Meierbachtol et al., 2016; Stearns and van der Veen, 2019) as many glaciers approach flotation conditions at the ice-ocean boundary. Thus, one explanation for the ongoing acceleration and retreat of outlet glaciers in Greenland, despite a pause in ocean thermal forcing (Wood et al., 2021), is the continued dynamic evolution of glaciers with sustained low basal drag conditions extending far inland.

## 5  Conclusions

We use the force balance method to examine the evolving stress fields for three neighboring glaciers with divergent retreat histories. Pre-retreat glacier dynamic changes and thinning are either small or non-existent; thus we suggest that these glaciers retreated as a result of change in processes at the ice ocean-boundary that occurred due to ocean warming. We find the stress state pre-retreat, largely a reflection of fjord-specific glacier geometry, uniquely determines how each glacier responds to ocean forced terminus retreat. Without terminus retreat the stress state is largely invariant, particularly for regions upstream of the terminus. Terminus retreat initiates pronounced changes in stress, but these changes are subsequent to retreat and are largely confined to the near-terminus region (within a stress coupling length of the terminus). For retreating glaciers, we find a drop in the fractional resistance to driving stress from longitudinal coupling after the onset of terminus retreat. This is followed by a temporal increase in lateral drag and associated glacier acceleration. For glaciers with low basal drag conditions near the terminus, acceleration leads to sustained retreat and greater mass loss in response to a period of climate forcing, e.g., Ingia Isbrae. For glaciers with regions of highly variable basal resistance upstream of the terminus, e.g., Umiamako Isbrae, terminus retreat is more readily stabilized. Together, these observations physically link the observed changes in glacier dynamics to ocean forced terminus retreat; explain the sometimes contrasting dynamics of neighboring glaciers around Greenland; and provide a mechanism for the sustained retreat of glaciers, despite a pause in ocean heat delivery to termini.

*Code and data availability.*  Processed ASTER data and merged data for figures are available at https://github.com/ecFlo/greenlandForceBalance along with scripts to make figures and calculate the force balance. All other data products used are publicly available.

*Author contributions.*  EC processed the data, wrote the ice flow model, and wrote the original manuscript. All authors designed the study, analyzed the results, and contributed to the final manuscript.

*Competing interests.*  The authors declare no competing competing interests.

*Acknowledgements.* We thank L. Girod, B. Csatho, and M. Weidersphan who provided assistance with DEM processing, M. Wood and D. Felikson for helpful discussions, A. Robinson for handling this manuscript, and L. Ultee and one anonymous reviewer for thoughtful comments that improved this manuscript. We acknowledge NASA Grant #80NSSC18K1477 and a JSG Fellowship to E. Carnahan for funding in support of this work.

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

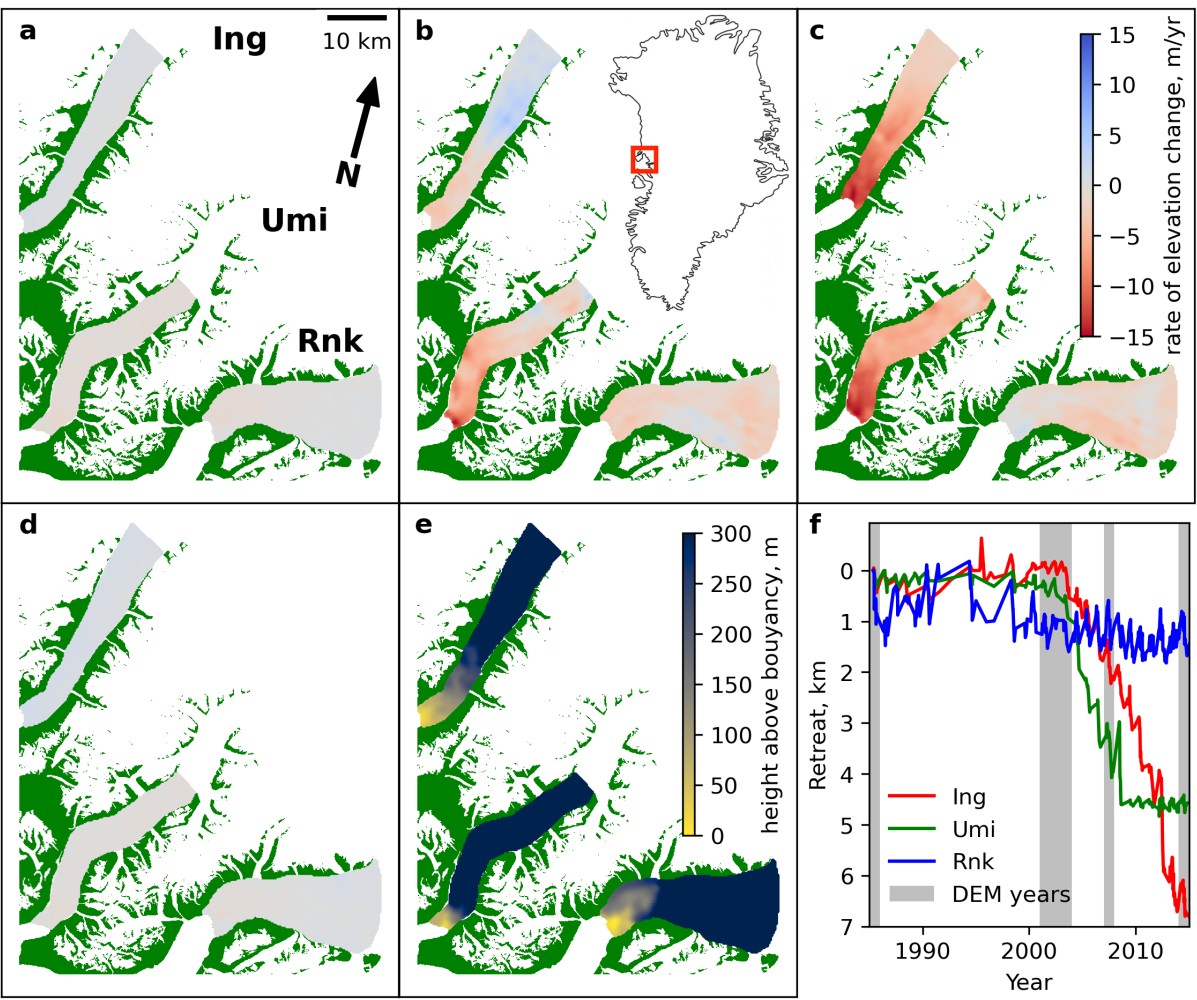

**Figure 1.** Rate of total elevation change for Ingia Isbrae (Ing), Umiamako Sermia (Umi), and Rink Isbrae (Rnk) from (a) 1985 to ~2002 (Ingia: 2003, Umiamako: 2002, Rink: 2001) (b) ~2002 to ~2007 and (c) ~2007 to ~2015. (d) Dynamic thinning, (total thinning minus thinning from surface mass balance prior to glacier retreat) from 1985 to ~2002. (e) Glacier height-above-buoyancy assuming an open connection to sea level, an ice overburden pressure upper bound, in 1985. In green is the land around the glaciers. All glaciers shown flow from the interior of Greenland on the right hand side of the figures to the ocean on the left where they terminate. (f) Glacier retreat histories since 1985 (Catania et al., 2018), along with dates of DEMs used.

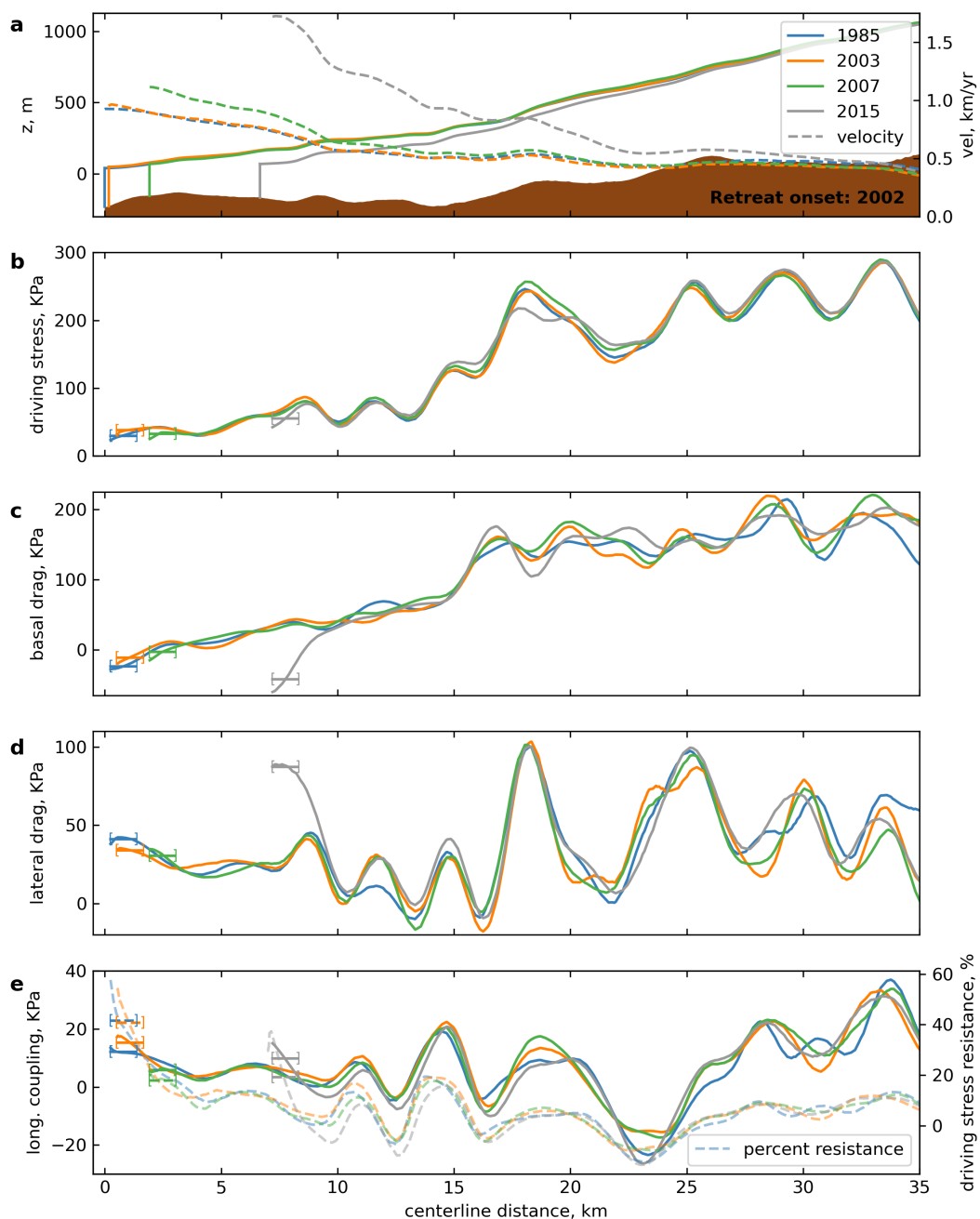

**Figure 2.** Time series of centerline dynamic components for Ingia Isbrae. Marked by the bracketed lines are the average values within one stress coupling length of the terminus at that time. (a) Surface and bed elevation and velocity profiles along flow. Along flow (b) driving stress, (c) basal drag, (d) lateral drag, (e) and longitudinal coupling and percentage of driving stress supported by longitudinal coupling. Positive values of driving stress act in the direction of flow, positive values of all other stresses oppose flow.

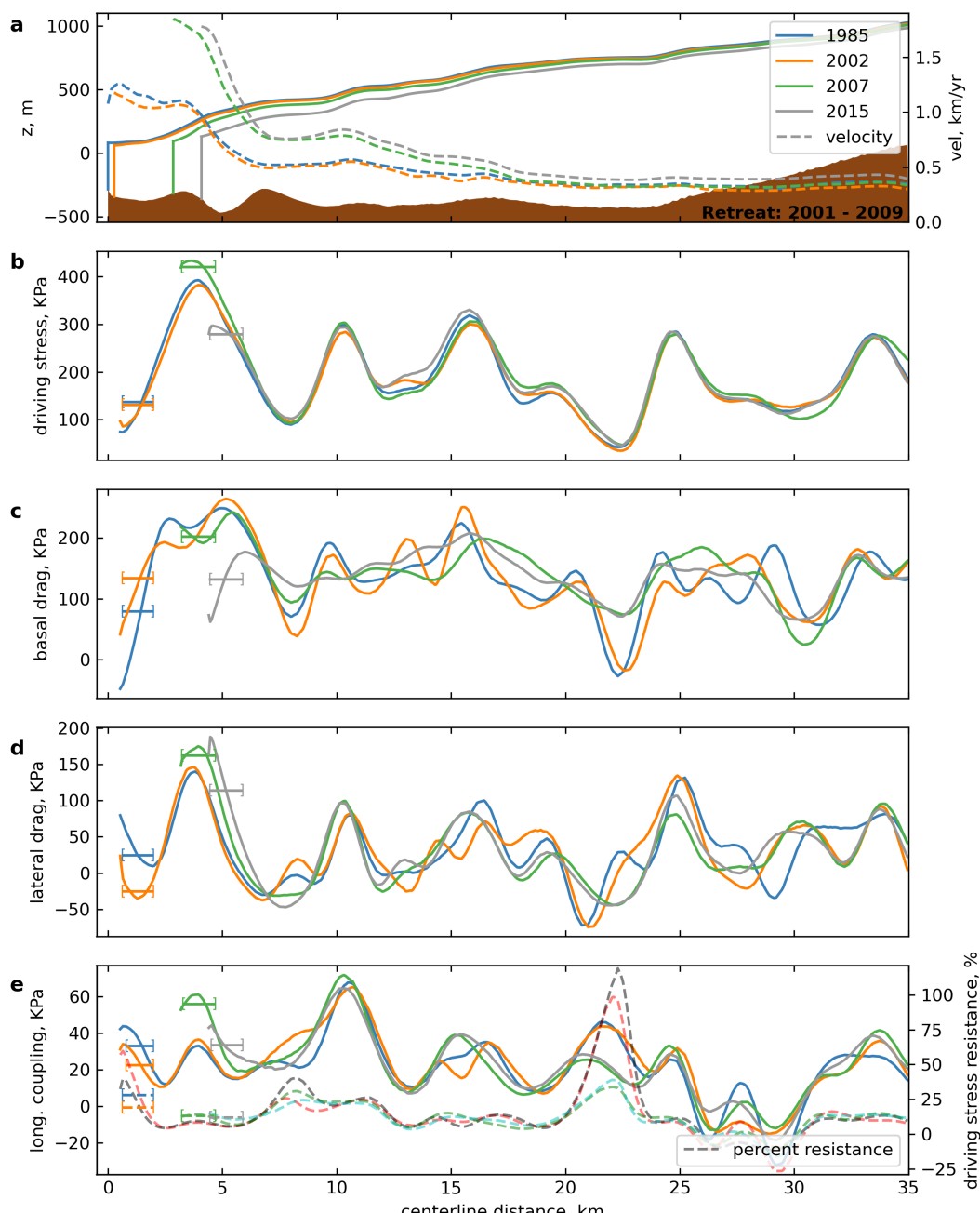

**Figure 3.** Time series of centerline dynamic components for Umiamiko Sermia. Marked by the bracketed lines are the average values within one stress coupling length of the terminus at that time. (a) Surface and bed elevation profiles, and along flow velocity. Along flow (b) driving stress, (c) basal drag, (d) lateral drag, (e) and longitudinal coupling and percentage of driving stress supported by longitudinal coupling. Positive values of driving stress act in the direction of flow, positive values of all other stresses oppose flow.

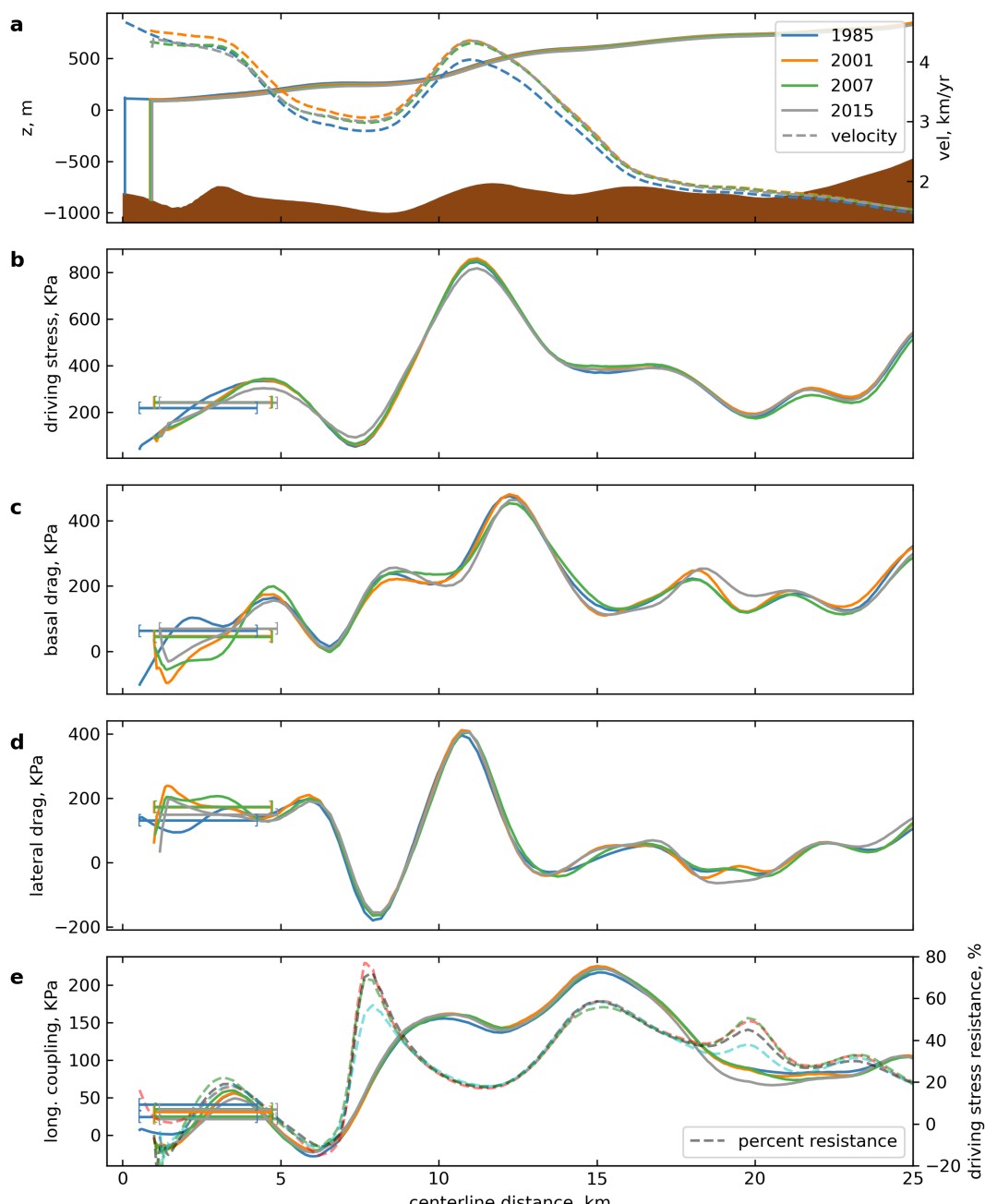

**Figure 4.** Time series of centerline dynamic components for Rink Isbrae. Marked by the bracketed lines are the average values within one stress coupling length of the terminus at that time. (a) Surface and bed elevation profiles, and along flow velocity. Along flow (b) driving stress, (c) basal drag, (d) lateral drag, (e) and longitudinal coupling and percentage of driving stress supported by longitudinal coupling. Positive values of driving stress act in the direction of flow, positive values of all other stresses oppose flow.