# Peer review of "Observed mechanism for sustained glacier retreat and acceleration in response to ocean warming around Greenland"

_The Cryosphere, 2022_

## Author Comment (AC1)

**Response to Reviewer 1**

**Summary**

This manuscript presents a stress balance analysis for three tidewater glaciers that have a contrasting retreat history over the study time period (1985-2015). By analysing the stress balance throughout a period of retreat, and by contrasting the glaciers, the authors aim to elucidate the possible drivers of retreat and controls on its ultimate duration.

The manuscript aims to make the significant point that at least for the 2 studied glaciers that retreated, it was a terminus perturbation that initiated retreat (so that retreat led to acceleration and then thinning), as opposed to for example a reduction in basal drag (for which the order would be acceleration then thinning then retreat). The manuscript furthermore shows how the pre-retreat stress configuration determines the susceptibility of a glacier to long-term retreat.

The manuscript is certainly important and is appropriate for The Cryosphere. I feel that there remains much uncertainty on the drivers and controls of tidewater glacier retreat and this manuscript can make a significant contribution to this topic. In addition it was a pleasure to read because the manuscript is well-written and the figures are excellent.

Having said all this, I do feel that three significant points need to be addressed before I feel completely convinced by the argument; these relate to (i) being more precise about the time stamping of the data and the implications for the analysis, (ii) a more thorough and extensive treatment of the errors and consideration of how these errors affect the interpretation of the results, and (iii) a more thorough analysis or discussion to discount thinning as a possible driver of retreat. I detail these points along with some more minor comments below.

Thank you for your interest in our study and for taking the time to review our paper. It is clear from your review that you took the time to understand the points we were making and offer a constructive and detailed critique. Prominent among the critique was a lack of consistent time stamping for glaciers around the year 2002 in the manuscript and the need for a more rigorous analysis of thinning induced retreat. We were able to make these changes and feel they have added to the clarity and persuasiveness of our findings. In addition, this reviewer suggested that to be fully convinced by the analysis, a better treatment of errors was necessary. We have formally propagated errors through our model and, when errors are commensurate to observed changes, discussed those changes in light of uncertainties. We feel that this review has substantially improved our manuscript.

**Major comments**

**Time stamping of data**

I found the time stamping of the data to be a bit imprecise and in some cases inconsistent. For example, on L87 and in Fig. 1 the date of the ASTER DEMs is stated as "∼2002", but in the figures and results the date of this DEM is sometimes said to be 2003 (e.g. legend on Figs. 2-4, L158). Could you be more precise at L87 about what " 2002" means? And then be consistent throughout the manuscript on this date?

Our apologies, and thank you for catching this. We mislabeled the year in Figs. 3-4. The year of the Ingia data is 2003 (i.e., why we stated it as 2003 on L158), the year of the Umiamako data is 2002, and the Rink data is 2001. We now on L87 explicitly state the months and dates of the data used for each glacier to represent the ∼2002 time period (Ingia: June 2003, Umiamako: July 2002, Rink: July 2001).

In relation to the timestamping of the DEMS, can the DEM (and therefore the accompanying stress analysis) really be said to be pre-retreat for Umiamako? On L160 you look at thinning between 1985 and 2002 but Umiamako began retreating in 2001 (L55). Does this affect your conclusions at all?

Umiamako and Ingia began retreating in 2002 and 2001 (Catania et al., 2018), but our datasets are from 2003 and 2002, respectively. ASTER DEMs were not available on the exact years of retreat for specific glaciers due to cloud cover, etc. One thing to mention is that this date of retreat is rough within a year or so of the retreat onset stated, see Umiamako Figure 2 in Catania et al., 2018. More importantly though, our main conclusion for the pre-retreat period is that there are negligible observed stress and thinning changes on Umiamako and Ingia prior to retreat. By analyzing data one year after retreat onsets, as opposed to one year before, the dataset is conservative for making this claim and therefore does not effects our conclusions.

What part of the season is the DEM appropriate for? I ask because if the seasonal surface elevation change from surface mass balance alone is significant (say 5m or something from late summer to late winter), then when you remove the surface mass balance in order to get the dynamic thickness change (L157) presumably it matters whether the DEM is timestamped to summer or winter. Perhaps this could affect the stress balance too. Similarly I note you used annual mean ice velocities – would seasonal variability in ice velocity (e.g. Howat et al. 2010) affect your results?

The ASTER DEMs (∼2002 datapoint) are all for summer months so they roughly maximize the amount of thinning that has occurred from 1985 to 2002, which is conservative for our findings of limited thinning prior to retreat (see year and months of DEMs now attached above and included in the paper). The 1985 DEM does not have a specific seasonal date to our knowledge. Although you are correct in stating that seasonality will affect the thickness, velocity is the major variable for year-to-year changes in stress, and, here, we consistently use annual velocity averages (O'Neel et al., 2005). To our knowledge seasonal velocities are not available/reliable in our study region before the early two thousands. Seasonal velocity changes are present for outlet glaciers in out study region, however changes are small <10 to 15% (Joughin et al., 2008) compared to the ∼50% increase in velocity we observe during retreat of Ingia and Umiamako. Although there are likely some smaller scale seasonal changes in stress associated with seasonal velocity changes super imposed on secular trends in stress, e.g., O'Neel et al., 2005; Enderlin et al., 2018, the aim of this study is to elucidate the year-to-year changes in stress during multi-year terminus retreat. For this purpose, we believe consistent use of annual velocities is best.

To sum up this point I feel the manuscript would be improved with a bit more clarity and explanation around the time stamping of the datasets, of how you have accounted for seasonal variability (or whether this matters), and how these details affect your results.

The reviewer pointed out an oversight of ours in mislabeling the year in Figs. 3 and 4. Furthermore, they raised the lack of specific dates for the DEMs for the years around 2002. We have added details above (and to the methods) and have provided a rationale for why the dates of DEMs used around 2002 do not effect, and are even conservative for, our conclusions. The discussion of seasonal changes is continued below, and will be mentioned in the manuscript discussion.

**Treatment of errors**

I found the treatment of errors to be a bit confusing and not sufficiently thorough. For example, on L108 it is stated that "errors in inferred basal drag using the force balance with BedMachineV4 are estimated to be <15 kPa", whereas on L114 you state a maximum error of 60 kPa in inferred basal resistance. Are these statements contradictory? In relation to the first statement, is it suggesting that the errors would be different with a different bed product?

Both values are from the analysis of Stearns and van der Veen, 2018. The less than 15 kPa on L108 is only for the uncertainty in basal drag that arises from the uncertainty in bed elevation, whereas the 60 kPa error stated on L104 is the estimate for the uncertainty that arises from all input data product errors, i.e., velocity, surface elevation, and bed elevation together; thus, the estimate is larger than 15 kPa. Both our study and Stearns and van der Veen, 2018 used BedMachine v3 thus we expect errors from bed topography alone to be similar. We made a mistake in referencing both as using BedMachine v4 on L108, that is now corrected to BedMachine V3.

In general, are you relying on Stearns and van der Veen (2018) for your error estimation as suggested by L114-115? But presumably your manuscript uses different input datasets (DEMs, velocities, updated BedMachine), and does different processing (the two sets of smoothing on L112 and L125), so aren't your errors likely to be different?

Although in most ways we use similar data products and smoothing methods to Stearns and van der Veen, 2018, and therefore we expect errors to be comparable to the ones they estimate (and find that to be the case), we understand that to be fully convincing it would be useful to have our own estimates of error for each glacier/year. Please see the reply below for the bulk of our analysis.

I also don't follow why the errors in inferred basal drag are necessarily "consistent in time" (L109). I can understand why this would be the case if an error is arising from your estimate of ice viscosity, and if ice viscosity is assumed to be constant in time. However, you have different inputs to your estimates at different times (DEMS, velocities) and so presumably these could give rise to different, time-variable errors?

We apologize for the confusion here, as we now see that this and your first comment in this section are likely linked to our poor wording. The invariance in time is just in reference to errors in stress arising from errors in bed topography. In our study, we assume that the bed elevation is invariant (stated in the previous line), thus the errors in stress components from errors in bed elevation are consistent in time. You are correct that errors in velocity (and surface DEMs) would be variable in time. We have now addressed this below.

Overall, I think it would be great if you could add to the methods a more thorough treatment and explanation of the errors, and I think it would be useful to add some shading or some sort of other indication of the error on the stress to Figs. 2-4.

We have now propagated errors through our model analytically following Taylor, 1997 and van der Veen, 2011, similar to previous work, e.g., van der Veen et al., 2013; Stearns and van der Veen, 2018. As we are primarily focused on relative changes in stress, and for the reasons stated above (invariance in time), we do not include uncertainty arising from errors in the bed DEM in our calculations. For lateral drag and longitudinal coupling, we estimate uncertainty due to errors in surface velocity datasets because (1) errors in velocity datasets vary in time, which results in relative changes in stress that our analysis focuses on and (2) velocity errors result in the vast majority of uncertainty in temporal stress changes, i.e., compared to surface elevation errors (van der Veen, 2013; O'Neel et al., 2005). We use values for velocity errors published alongside the dataset used here (Gardner et al., 2019) and analytically propagate them through our model to calculate stress uncertainty. We assume that errors in annual velocity maps and stress fields arise independently, i.e., spatial averaging during resampling reduces error. It is important to note that the values given for error by Gardner et al., 2019 are noted to "allow for the formal propagation of errors" but "provided errors... should be used as qualitative metrics for assessing errors." As a result, the formal propagation of velocity errors through our model should be taken as a similarly qualitative assessment, albeit the best one available to us at present.

The results of the uncertainty quantification in the form of the shaded plots you suggested are at the end of this response to reviewer comments (Figs. 1-3) and will be added to the supplementary materials in the manuscript. The method for estimating uncertainty will be added to an adapted methods section. Furthermore, in regions/years

where uncertainty is commensurate to the stresses reported, the results and conclusions will be discussed in light of that uncertainty (next comment below). During certain years we find some regions of very high uncertainty due to small regions of coincident low strain rates and high velocity errors. However, on average we find comparable uncertainty values for stress terms to previous studies in Greenland, e.g., Enderlin et al., 2016; Stearns et al., 2018.

Lastly, and depending on your response to the above points, I think it would be great to be more conscious of the errors when discussing the results. Two particular places I feel this could be important are: (i) on L167 when you talk about "a drop in longitudinal coupling resistance in the near-terminus region of 10 kPa" – is this outside of uncertainty?, and (ii) on lines 157-163, could you comment a bit on what the errors are on these dynamic thickness changes? Ideally you would have a +/- attached to each estimate. I ask because these are relatively small changes that are comparable to the uncertainties you describe in L91-93, and furthermore you have removed a RACMO surface mass balance signal that presumably itself has significant uncertainty. Therefore, can we say that these glaciers are dynamically thinning or thickening outside of uncertainty?

Your feeling on this was founded and although largely the estimation of uncertainty did not fundamentally change our findings, on (i) the change we report is commensurate to the uncertainty we calculate. Therefore, that change is now only discussed in context of the present uncertainty in our methods. For (ii) you are also correct and these changes, e.g., 10 m of thinning/thickening, is within the RMS error in DEM accuracy we reported earlier in the manuscript. However, our claim is that we observe largely negligible changes in elevation pre-retreat, therefore in contrast to (i) including uncertainty largely strengthens our claim. Another important place to include uncertainty estimates was on L138, but similar to (ii) it strengthens the points made. All of these changes are now discussed in context of the underlying uncertainty.

**Thinning versus terminus perturbation-induced retreat**

One of the principal take-homes from the manuscript is that the retreat of Ingia and Umiamako is initiated by a perturbation at the terminus, because little thinning is observed prior to retreat (e.g. L162). I largely agree, but if we are going to be really rigorous, I feel we should ask what threshold of thinning we consider to be insufficient to

drive retreat. For Umiamako, there is some thinning of 5-10 m prior to retreat (if we take the 1985-2002 time period to be prior to retreat – see above). Clearly this is a small amount of thinning relative to what you get once full-on retreat is initiated, but that doesn't completely rule it out as the perturbation that started the retreat. Is the height above buoyancy for Umiamako (Fig. 1e 3a) such that 5-10 m of thinning could unground a significant portion of the terminus? Based on Fig. 3a it does look like the bed deepens inland in the first few km such that the glacier might be approaching flotation there in 2002/2003. I feel that a bit more quantitative analysis and discussion is needed here to fully back up the idea that thinning prior to retreat is not the driver of retreat. Perhaps adding a plot of height above buoyancy along the flowline in the near terminus region would help?

Continuing from your previous comment on uncertainty, the observed changes in thickness pre-retreat were largely negligible, or potentially within error. However, you are correct it is possible that ∼10 m of thinning estimated on Umiamako resulted in thinning to flotation and retreat. One way to estimate how much retreat could be induced from thinning alone is to follow Wood et al., 2021 and Thomas and Bentley, 1978. Here, the thinning induced retreat rate is calculated as,

$$q_s = dh/dt[(1 - \rho_w/\rho_i)\beta_s - \alpha_s] \tag{1}$$

where $dh/dt$ is the thinning rate, $\beta_s$ is the basal slope at the ice front, $\alpha_s$ is the slope of the glacier surface at the ice front, $\rho_i$ is the density of ice, and $\rho_w$ is the density of water. Following Wood et al., 2021, if $dh/dt$ is positive than $q_s = 0$. So, for Ingia, which thickened, $q_s = 0$. For Umiamako, $dh/dt = 10/17$ m/yr. Integrated over $\delta t = 17$ year period from 1985 to 2002, $q_s\Delta t$, results in just 0.5 m of retreat. Thank you for the suggestion to make this claim more rigorous, we will include the calculation in the manuscript and, although we both heuristically thought this was a negligible amount of thinning, we feel the additional calculation strengthens our conclusion.

**Minor comments**

L23-25 – This may be preference, but I feel this sentence would read better if all the references were put at the end.

Done.

L70 - Could you add a bit more detail (particularly including equations) for how you calculate the resistive stresses from the velocities? I see that you have written it in words around L70 but for clarity it would be great to see "R_xx = . . ."

The resistive stresses neglecting vertical shearing are given by

$$R_{xx} = B\dot{\varepsilon}_e^{1/n-1}(2\dot{\varepsilon}_{xx} + \dot{\varepsilon}_{yy}), \tag{2}$$

$$R_{yy} = B\dot{\varepsilon}_e^{1/n-1}(\dot{\varepsilon}_{xx} + 2\dot{\varepsilon}_{yy}), \tag{3}$$

$$R_{xy} = B\dot{\varepsilon}_e^{1/n-1}\dot{\varepsilon}_{xy}, \tag{4}$$

where $B$ is the viscosity rate factor, $\dot{\varepsilon}_{ij}$ is the surface strain rate and the effective strain rate is given by

$$\dot{\varepsilon}_e = (\dot{\varepsilon}_{xx}^2 + \dot{\varepsilon}_{yy}^2 + \dot{\varepsilon}_{xx}\dot{\varepsilon}_{yy} + \dot{\varepsilon}_{xy}^2)^{1/2}. \tag{5}$$

These equation will be merged into the methods of the main text. See van der Veen, 2013 ch. 11 for more details.

L104 – can you clarify whether "all years in our study period" means every year from 1985-2015 or just 1985, 2002, 2007 and 2015?

Just the years in our analysis, i.e., 1985, 2001 (Rink), 2002 (Umiamako), 2003 (Ingia), 2007, and 2015.

L105 – putting the last part in parenthesis might be better grammatically?

Done.

Fig. 2 caption – suggest "terminus at that time" would be better than "current terminus".

That's more clear. Done.

Fig. 1d could possibly benefit from a different color scale because it's difficult for the reader to evaluate the statements in L157-163 when much of the glacier looks to have a surface elevation change of approximately 0.

We see your point, however, we use the same colorbar to show the thinning rate for all years, which allows for easy year-to-year comparison. Largely our main point for this time span, 1985 to ∼2002, is there is near-uniform negligible (compared to post retreat) thinning (Fig. 1a) and dynamic thinning (Fig. 1d), thus the lack of observable change and the slight changes we can see is a feature of the plot.

L163 – "ice-ocean processes" – I'm wondering if perhaps this should more accurately be "calving front processes", because I guess in theory something like increased calving driven by hydrofracture would be consistent with your observations but is not an ice-ocean process.

We have made the change to your more general term, as the process you identify is a potential mechanism.

L165 – "along-flow gradients in longitudinal stresses support driving stress" – is the "gradients" part necessary here? Wouldn't it just be longitudinal stresses supporting driving stresses?

You are correct. We have removed gradients.

L176-177: "The Ingia terminus region does not experience a significant change in driving stress or basal drag even as the terminus retreats" – in Fig. 2c the basal drag does change significantly. Although the values do become unphysical, I feel that this statement needs modified. Presumably since all stresses must sum to 0, the negative basal drag values are telling us that one of the other stress components is slightly out too?

That sentence you quote is meant to refer to the period up to 2007, i.e., "...drag as the terminus begins to retreat" and the following sentence refers to the 2015 period, where we will now also discuss the drop in basal drag that occurs during the increase in lateral drag from 2007 to 2015. The original wording did not make that clear, thus we have improved the wording here to aid understanding.

L208 – "compressional flow" – to me, compression, at least in the along-flow direction, is when the velocity is decreasing along-flow, which would be between 8-11 km on Fig. 4a. I don't really follow how "the compressional regime is evidenced by large gradients in longitudinal stress" because this does not follow my understanding of compressional flow. If we were only consider the along-fjord longitudinal stress (ignoring the across-glacier direction) then wouldn't compressional flow be associated with negative longitudinal stresses rather than gradients in longitudinal stresses? Or put another way, a flow can still be entirely extensional even in the presence of gradients in longitudinal stresses. Perhaps the authors can expand/explain?

We have cleaned up the language to only say that the driving stress is highly supported by longitudinal coupling throughout the trough for Rink (what is shown in our results) and do not comment on the compressional or tensional aspect of the ice flow.

L213 – on L127 you state that "calculated stresses can not be interpreted at length scales below the stress coupling length of each glacier" while here you state that "the proportion of the driving stress supported by longitudinal resistance increases upstream of the terminus region". In making this statement are you not interpreting the stress at length scales below the stress coupling length? The increase in longitudinal resistance looks to be within the stress coupling length, and then further upstream the longitudinal resistance decreases for a bit.

You raise a good point here. We double checked the veracity of our claim that the proportion of longitudinal coupling that supports driving stress increases up glacier for Rink and that this trend is present above the length scales of the stress coupling length. We calculate the average of longitudinal coupling over a 3.7 km window (the stress coupling length of Rink) for 1985 and plot the value of the average stress at the midpoint of the window (Figure 4 below). We find that the increase is less than we originally observed and the increase in percentage resistance to driving stress up glacier is a mere few percents within 2 km behind the SCL region, or near the measurement error of our methods. We thus have softened this claim substantially. We now state that Rink merely appears to not have a pattern of decreasing longitudinal resistance to driving stress within two kilometers of the near-terminus region and thus a small scale retreat would not result in the same loss of longitudinal coupling resistance that it does for Ingia and Umiamako. We save any discussion of the possible mechanisms of stability for Rink to the discussion where we talk about it in context of Wood et al., 2021, which finds Rink has a fjord morphology that limits access of warm ocean waters, L222.

L240 – perhaps "little resistance from basal drag" would be more appropriate since it is non-zero.

Done.

L237-249 or another relevant place – I wonder if you could discuss to what extent your findings on how the geometry/stress state determine the susceptibility to retreat

relate to those of Felikson et al. (2021)? I feel the work of Felikson et al. (2021) should be referenced and discussed in relation to your results.

We would have loved to add more about this work here, however, as far as direct comparison, only the Ingia glacier area in our study contains the Peclet limit (Peclet running maxima of three), ∼15 km from the 1985 terminus (Felikson et al., 2017). For Umiamako and Rink the Peclet limit is >40 km inland (Felikson et al., 2017). This lack of ability to analyze more than one glacier largely precluded us from anything but basic comparisons. For Ingia, the flow regime from the terminus to the Peclet limit is characterized by low driving stress and basal drag. At the Peclet limit both driving stress and basal drag increase by nearly a factor of two. This suggests that the ability for thinning waves to diffuse up glacier is linked to the stress state of the glacier and potentially the ability for stress changes to be transferred upstream (Bondizo et al., 2017). Fundamentally, both the pre-retreat stress state that we identify and the Peclet thinning limit identified by Felikson et al., 2017 highlight the importance of the glacier geometry in determining the dynamic response to retreat. Furthermore, we find here that thinning is subsequent and in response to retreat. This ordering is consistent with, and helps to fill in, the chain of events suggested by Felikson et al., 2017. We will add these points to the discussion as well as highlight the interesting but somewhat anecdotal results we find for the Ingia Peclet limit to motivate further study.

L250-265: Relating to the wider significance of their results, I wonder if the authors could include a short discussion on what sets the basal drag? From a subglacial hydrology perspective, one might expect basal drag to depend on conditions at the bed – i.e. the presence of water or sediment and the state of the hydrological system. On the other hand, in your results (Figs. 2-4), there is a strong imprint of the overall stresses on the basal drag, and the overall stresses respond to the geometry of the bed and glacier, suggesting that the large-scale geometry of the system plays a role in setting the basal drag. How do we reconcile these two different viewpoints? This feels important because how the system responds in the future might be different depending on what is setting the basal drag. I understand that addressing this properly is beyond the scope of your results, but I think some discussion would be very helpful.

In this contribution, we find a strong role of geometry in setting the basal drag for outlet glaciers in Greenland. This finding is largely due to the long time scales over which

we analyse changes and the focus on relative changes in stress, however, no friction law would be complete without taking into account subglacial hydrology and bed materials (Joughin et al., 2019). There are clear examples where subglacial hydrology is paramount. For example, on seasonal timescales the geometry is only slightly different and yet glacier dynamics show substantial variability that is best explained by changes in subglacial hydrology (Howat et al. 2010). Such changes are far below the temporal resolution of our analysis, but provide a potential avenue for future work to decipher where and when either of the two controlling factors you mention is dominant. We feel that our work largely elucidated geometric influences on basal drag due to the course temporal resolution of our analysis and our focus on outlet glacier areas in Greenland dominated by basal sliding on soft beds (see for example, Andrews et al., 1994). Our discussion already includes a portion on the importance of geometry in glacier dynamics and the response to retreat, which will be expanded by the comparisons to Felikson et al., 2017. Now, we will also mention the alternative subglacial hydrology perspective and the strong evidence for it in setting basal drag and glacier dynamics more broadly, e.g., Howat et al., 2010; Zwally et al., 2002; Schoof, 2005.

L279 – "reductions in heat delivery to termini" – I feel this needs some qualifying or further specification. I guess that you are probably referring to the Wood (2021) paper, but as written I feel there is a danger of someone thinking that in general we expect reductions in ocean heat delivery in the future. This also applies to L11.

We see your point and, yes, we were referring to Wood (2021). Instead of reductions, we now refer to it as a "pause" in ocean heat delivery to termini. We will make a similar change in L11.

**Typos**

L61 – no need for parenthesis on reference

L109 – "affect" rather than "effect"

Thanks! Done.

**Literature cited**

Felikson, D., Catania, G. A., Bartholomaus, T. C., Morlighem, M., Noël, B. P. Y. (2021). Steep glacier bed knickpoints mitigate inland thinning in Greenland. Geophysical Research Letters, 48, e2020GL09011

Gardner, A. S., M. A. Fahnestock, and T. A. Scambos, 2019 [update to time of data download]: ITS_LIVE Regional Glacier and Ice Sheet Surface Velocities. Data archived at National Snow and Ice Data Center; doi:10.5067/6II6VW8LLWJ7

Andrews, J. T., Milliman, J. D., Jennings, A. E., Rynes, N., Dwyer, J. 1994. Sediment Thicknesses and Holocene Glacial Marine Sedimentation Rates in Three East Greenland Fjords (ca. 68N). The Journal of Geology, 102(6), 669–683. http://www.jstor.org/stable/30065642

Taylor, J.R. 1997. An introduction to error analysis: the study of uncertainties in physical measurements. Second edition. Sausalito, CA, University Science Books.

[Figure]

**Figure 1.** Time series of centerline stress components for Ingia Isbrae along with qualitative uncertainties in stresses from errors in velocity data. Along flow (a) driving stress, (b) basal drag, (c) lateral drag, (d) and longitudinal coupling. Positive values of driving stress act in the direction of flow, positive values of all other stresses oppose flow.

[Figure]

**Figure 2.** Time series of centerline stress components for Umiamako Isbrae along with qualitative uncertainties in stresses from errors in velocity data. Along flow (a) driving stress, (b) basal drag, (c) lateral drag, (d) and longitudinal coupling. Positive values of driving stress act in the direction of flow, positive values of all other stresses oppose flow.

[Figure]

**Figure 3.** Time series of centerline stress components for Rink Isbrae along with qualitative uncertainties in stresses from errors in velocity data. Along flow (a) driving stress, (b) basal drag, (c) lateral drag, (d) and longitudinal coupling. Positive values of driving stress act in the direction of flow, positive values of all other stresses oppose flow.

[Figure]

**Figure 4.** Plot of Rink Isbare (top) longitudinal coupling and (bottom) longitudinal coupling resistance to driving stress in 1985 averaged over the stress coupling length of Rink (3.7 km) and plotted at the midpoint of the averaging window.

---

## Author Comment (AC2)

**Response to Lizz Ultee**

Carnahan, Catania  Bartholomaus present a stress-balance analysis of three neighboring outlet glaciers in Greenland: Ingia Isbrae, Umiamako Isbrae, and Rink Isbrae. Despite their proximity, the three outlets showed a variety of retreat histories in recent decades, making them a useful site for comparative analysis. The study is interesting and relevant, certainly worth publishing in The Cryosphere. In some places, I suggest revising to make the text more precise. The discussion should also be strengthened, especially with regard to the Greenland-wide implications of the work. Finally, I suggest the authors take another editing pass through the text to simplify the language.

Thank you for taking the time to review our paper, understanding the arguments made, and making comments that will be helpful for contextualizing our results. This review largely focused on a few of the more sweeping claims we made in the discussion and some unclear statements throughout. We have cleaned up the language in the text and have spent time qualifying and specifying the two broad claims made in the discussion. We now feel that the discussion is supported by our results. This review helped us to present the potential broader implications of our results in a rigorous way. Thank you.

**Specific comments**

L58: would it be accurate to say simply "no secular trend emerges from seasonal fluctuations"?

Yes, thank you.

L62-65: Please revise here to describe why the force balance method is suitable for your study. Stating which studies have used it before does not provide your scientific motivation for using it here.

We have now added the motivation for this method, i.e., this method provides snapshots in time of the glacier stress state and is therefore useful to examine how the stress state varies in time during changes in glacier terminus position.

L84: I would not cite the Minchew et al. 2019 comment as evidence of "uncertainty" in the basal sliding relation. Minchew et al. 2019 do not advocate for or provide evidence supporting any sliding relation different from the one tested by Stearns and van der Veen

2018. If the authors wish to highlight the longstanding debate about an appropriate form of the sliding relation, I suggest older (classic) references: Kamb 1970; Budd et al 1979; Weertman 1957, 1964, 1972; Lliboutry 1968, 1975, 1979; Nye 1969, 1970, etc. No need to cite them all, of course, but to me the longstanding and ongoing debate is more compelling than the "present uncertainty".

We appreciate your suggestions and have removed the Minchew et al., 2019 reference and added references to Kamb, 1970; Nye, 1970; and Lliboutry, 1979.

L129-130: Please describe how you calculated the stress coupling length, for readers who are not familiar with it.

We estimate the stress coupling length (SCL) following Enderlin et al, 2016 as $SCL = 4H$, where $H$ is the average ice thickness in the region of interest. We will add the calculation to the manuscript.

L137: "Average absolute changes in inferred basal drag..." - What does this average mean? Is it the average of per-point change from one time step to the next, at each point along the flowline? Or is it averaged in some other way?

Yes, it is the average of per-point change from one time step to the next. We will change these lines in light of the new calculations of uncertainty and will make the description of the average more specific.

L142-143: "...implies that in the absence of terminus retreat glacier dynamics are largely invariant" can you be more specific? Invariant in what way? Which observable variables would you expect not to change, and over which time scales?

We have made the statement more specific to our observations. "...implies that in the absence of retreat secular glacier dynamics are largely invariant for these three glaciers during our study period." This was clearly too broad as changes in subglacial hydrology, surges, etc. would all be examples of changing glacier dynamics that occur without retreat. Thanks for pointing this out.

L154-155: "The climate system can also force retreat through processes at the ice-ocean boundary (Motyka et al., 2011)." Is the backstress example (3) from Nick et al. 2009 in the preceding sentence not an example of forcing at the ice-ocean boundary? Please clarify wording in these sentences.

Although you are correct, we feel there is a subtle distinction here that we did not previously make clear. In Nick et al., 2009 they trigger retreat by changing backstress, that stress is triggered at the ice-ocean boundary, as you point out, but retreat is subsequent and caused by the stress change, as opposed to increased frontal ablation that causes retreat then stress changes and acceleration. We acknowledge that these processes are clearly coupled, we try and show that in our paper, and our distinction is somewhat pedantic. To tie them closer we now state, "3) decreases in terminus backstress forced by ice-ocean interactions that causes acceleration and subsequent retreat" and in the below line say "Similar to 3), the climate system can also force retreat through increased frontal ablation and successive dynamic changes (Motyka et al., 2011)." The reason for being nit picky here is that we observe that stress changes largely occur subsequent to, and because of, terminus retreat.

L202-203: would it be more accurate to say "Ingia experienced an acceleration in ice flow velocity and associated two-fold temporal increase in lateral drag"?

Yes - thanks!

L234-236: "If these observations are representative of the ice sheet as a whole..." is a big assumption. Greenland has more than 200 outlet glaciers, and they are quite heterogeneous in terms of geometry, surface climate, ocean access, etc.—and I don't mean to lecture the authors on this, as I know they have published papers on the heterogeneity of Greenland outlet glaciers. Anyway, I suggest toning down this generalization, or else including several more sentences of interpretation.

We acknowledge as written the claim is speculative. We have attempted to tone down this generalization. We now say, "These observation suggest that one potential mechanism for the widely observed acceleration of outlet glaciers around Greenland (Murray et al., 2014; King et al., 2018, King et al., 2020) is a response to coupled changes in lateral drag and near-terminus longitudinal backstress initiated by terminus retreat. However, glaciers around Greenland inhabit a wide range of geometries, climate regimes, and fjord geometries (Morlighem et al., 2017; Catania et al., 2021; Felikson et al., 2021) so future study is likely necessary to understand the prevalence of the proposed dynamic connection between retreat and acceleration."

L239: can you specify \*what\* about the fjord geometry permits low basal drag extending inland?

The region of low basal drag occurs where the submarine bed topography is shallowly retrograde extending far inland to 15 km. We will added this to L239 in the discussion.

L261-262: please look for a few more references to support the claim that low basal drag conditions "are not restricted to these well-studied glaciers, but occur around Greenland". Shapero et al (2016) considers only the three outlet glaciers that I would argue are the most well-studied: Kangerlussuaq, Helheim, and Sermeq Kujalleq. That is not sufficient support for the more general statement that follows in L262-264. It would be very interesting to know how generalizable your findings are to other outlets, but we need evidence from more outlets than the "Big Three" for that generalization.

We have added more references here, removed the reference to "well-studied", softened the general claim in the following sentence, and backed up the generalization with a mechanistic explanation for low basal drag conditions at or near glacier termini in Greenland, i.e., "Conditions of low basal drag throughout the near-terminus region of glaciers are not restricted to the glaciers in this study, but occur around Greenland (Shapero et al., 2016, Sergienko et al., 2014, Seddik et al., 2018, Bartholomaus et al., 2016, Nick et al., 2012, Meierbachtol et al., 2016, Stearns et al., 2019) as many glaciers approach flotation conditions at the ice-ocean boundary. Thus, one potential explanation for the ongoing acceleration and retreat of outlet glaciers in Greenland, despite a pause in ocean thermal forcing (Wood et al., 2021), is the continued dynamic evolution of glaciers with sustained low basal drag conditions extending far inland."

Abstract lines 9-11: This claim is related to the manuscript, but not supported by the evidence you present. See above. Please remove, rephrase, or provide more evidence in the main text.

The last sentence of the abstract was re-written to align more closely with the evidence presented, i.e., "Glaciers with similar basal stress conditions occur around Greenland. Our results suggest that for such glaciers, dynamic mass loss can be sustained into the future despite a pause in ocean forcing."

I was surprised not to see any discussion of this manuscript's findings in context with those of Felikson et al. (2017, 2021), especially given the overlap in authorship. It would be helpful to me as a reader if the authors would discuss those works.

We would have loved to add more about this work here, however, as far as direct comparison, only the Ingia glacier area in our study contains the Peclet limit (Peclet running maxima of three), ∼15 km from the 1985 terminus (Felikson et al., 2017). For Umiamako and Rink the Peclet limit is >40 km inland (Felikson et al., 2017). This lack of ability to analyze more than one glacier largely precluded us from anything but basic comparisons. For Ingia, the flow regime from the terminus to the Peclet limit is characterized by low driving stress and basal drag. At the Peclet limit both driving stress and basal drag increase by nearly a factor of two. This suggests that the ability for thinning waves to diffuse up glacier is linked to the stress state of the glacier and potentially the ability for stress changes to be transferred upstream (Bondizo et al., 2017). Fundamentally, both the pre-retreat stress state that we identify and the Peclet thinning limit identified by Felikson et al., 2017 highlight the importance of the glacier geometry in determining the dynamic response to retreat. Furthermore, we find here that thinning is subsequent and in response to retreat. This ordering is consistent with, and helps to fill in, the chain of events suggested by Felikson et al., 2017. We will add these points to the discussion as well as highlight the interesting, but somewhat anecdotal, results we find for the Ingia Peclet limit to motivate further study.

Technical corrections

L24-25: "heterogeneous changes in elevation (Csatho et al., 2014; Felikson et al., 2017) and velocity (Moon et al., 2020)."

L25: "This means..." - what is "this"?

L37: "circumnavigate elevation data scarcity" —¿ "circumvent scarce elevation data"

L61 replace parenthetical citation with in-text citation

L78-80: "Such observations... (Shapero et al 2016)" — I suggest removing this sentence. I want to know more about what you did, not necessarily what others have done before.

L84: "Zoet and Science, 2020" —¿ "Zoet and Iverson, 2020"

L103: "two additional data products are necessary..." should have a colon rather than a comma

L116: "model's" missing apostrophe

L138: why not reference the relevant figure directly after mentioning each glacier? That is, "16 kPa for Rink (Fig. 4); 25 kPa for Umiamako (Fig. 3); and 11 kPa for Ingia (Fig. 2)"

L147: "buoyant"-¿ "buoyancy"

L150: remove semicolon

Sections 3.2-3.4: check verb tenses. "We observe" in the present tense makes sense to me, but in describing the results you use both past and present tense. For example, "Umiamako experiences...driving stress substantially increased" appear together in one sentence.

L184: consider "not only lateral drag" rather than "just"

L244: "maxima" is plural. Try "A maximum...is" or "Maxima...are".

L250: colon rather than comma when starting the list of glaciers with observed or modeled stress fields

L251-254: please use the official name of Greenland's largest outlet: Sermeq Kujalleq. For clarity, you might consider "Sermeq Kujalleq (also called Jakobshavn Isbrae)" or similar. See Bjørk, Kruse Michaelsen (2015).

L269: "dictates"-¿ "determines"

Figure 1: Please annotate a bit more and/or include more description in the caption. For example, what do the green and white regions indicate on each plot? Can you include arrows to show the direction of ice flow on one of the plots? I suggest glossing the abbreviations you use (e.g. "Ingia (Ing)") in the caption, even if you think they are obvious.

All fixed, clarified, or expanded on - thank you!

**References in this review**

Felikson, D., Bartholomaus, T., Catania, G. et al. Inland thinning on the Greenland ice sheet controlled by outlet glacier geometry. Nature Geosci 10, 366–369 (2017). https://doi.org/10.1038/ngeo2934

Felikson, D., Catania, G. A., Bartholomaus, T. C., Morlighem, M., Noël, B. P. Y. (2021). Steep glacier bed knickpoints mitigate inland thinning in Greenland. Geophysical Research Letters, 48, e2020GL090112. https://doi.org/10.1029/2020GL090112

Bjørk, A. A., Kruse, L. M., and Michaelsen, P. B.: Brief communication: Getting Greenland's glaciers right – a new data set of all official Greenlandic glacier names, The Cryosphere, 9, 2215–2218, https://doi.org/10.5194/tc-9-2215-2015, 2015.

---

## Author Response (AR2)

**Response to the Editor**

To the authors,

The manuscript is very well written and the message is clear and interesting. Furthermore, you have thoroughly addressed all reviewer comments. I only have some additional minor suggestions below before proceeding with publication.

Best regards, Alex

Thank you for your handling of our manuscript, your helpful comments, and your interest in our study! With your help, we feel we have now addressed any lingering issues.

Specific comments:

L4: precede retreat => preceding retreat

Done. L4

L49: "elucidate dynamic changes" is not very precise (and "elucidate" is repetitive with start of paragraph). Consider rephrasing for clarity. What do the inverse methods produce more explicitly?

We have made this statement more specific to the utility of inverse methods in estimating stresses for our study. L49-50

L75: "surface strain rate" <= This should rather be clarified by another sentence. In this depth-integrated formulation, this should represent the depth-averaged strain rate, correct? Then, since vertical shearing is neglected, it could be stated that the surface strain rate is equal to the depth-averaged strain rate, which allows the link to the observations. I think being explicit here is more appropriate.

Yes, you are correct. We have added an additional sentence clarifying this point. L75, L78-79

L135: following Taylor (1996); van der Veen, C.J. (2013) => following Taylor (1996) and van der Veen, C.J. (2013)

Done. L136

L164: (Fig. 2 and 3); coupled => (Fig. 2 and 3). Coupled

Done. L165-166

L165-166: implies that <= this implies that

Done. L166

L192: retreat => retreat,

Done. L193

L202: Umi => Umiamako

Done. L203